# Classification of soybean seeds based on RGB reconstruction of hyperspectral images

**Xu Yang[1], Kejia Ma[1], Dejia Zhang[1], Shaozhong Song[ID][2]\*, Xiaofeng An[2]**

**1** College of Electronic Information Engineering, Changchun University of Science and Technology, Changchun, China, **2** Jilin Engineering Normal University, Changchun, China

\* songsz@jlenu.edu.cn

**Data Availability Statement:** https://github.com/5120191452/SENet-ResNet34-DCN/blob/Datasets/SENet-ResNet34-DCN.zip

**Funding:** 1. the Natural Science Foundation of Jilin Province (No.2020122348JC)

## Abstract

Soyabean is an incredibly significant component of Chinese agricultural product, and categorizing soyabean seeds allows for a better understanding of the features, attributes, and applications of many species of soyabean. This enables farmers to choose appropriate seeds for sowing in order to increase production and quality. As a result, this thesis provides a method for classifying soybean seeds that uses hyperspectral RGB picture reconstruction. Firstly, hyperspectral images of seven varieties of soybean, H1, H2, H3, H4, H5, H6 and H7, were collected by hyperspectral imager, and by using the principle of the three base colours, the R, G and B bands which have more characteristic information are selected to reconstruct the images with different texture and colour characteristics to generate a new dataset for seed segmentation, and finally, a comparison is made with the classification effect of the seven models. The experimental results in ResNet34 show that the classification accuracy of the dataset before and after RGB reconstruction increases from 88.87% to 91.75%, demonstrating that RGB image reconstruction can strengthen image features; ResNet18, ResNet34, ResNet50, ResNet101, CBAM-ResNet34, SENet-ResNet34, and SENet-ResNet34-DCN models have classification accuracies of 72.25%, 91.75%, 89%, 88.48%, 92.28%, 92.80%, and 94.24%, respectively.SENet-ResNet34-DCN achieves the greatest classification accuracy results, with a model loss of roughly 0.3. The proposed SENet-ResNet34-DCN model is the most effective at classifying soybean seeds. By classifying and optimally selecting seed varieties, agricultural production can become more scientific, efficient, and sustainable, resulting in higher returns for farmers and contributing to global food security and sustainable development.

## 1. Introduction

Soybean is an annual herbaceous plant of the Soybean genus in the Leguminosae family that originated in China and has been cultivated for over 5000 years. It is now extensively cultivated across the country and is produced in the northeast, north China, Shaanxi, Sichuan, and the lower reaches of the Yangtze River, and is farmed more in the Yangtze River Basin and the southwest of the nation, with the highest grade soybeans in the northeast [1]. Soybean is not only delicious but also highly healthy [2]. Diverse soybean varieties may have considerable

**Competing interests:** The authors declare that they have no known competing financial interests or personal relationships that could have appeared to influence the work reported in this paper. The authors have declared that no competing interests exist.

yield variances; some have been selected to increase productivity, while others may have more tolerance and flexibility to generate higher yields under a variety of environmental conditions. Hundreds of distinct soybean varieties have been developed as technology has advanced and breeding procedures have become more diverse. However, some soybean cultivars produced in the same place have such similar appearances that they are impossible to discern with the human eye. At the same time, certain poor seeds on the market are difficult to identify, and they have fewer nutritional qualities, including proteins, carbs, lipids, vitamins, and minerals. As a result, precise soybean variety categorization has become increasingly vital.

Hyperspectral imaging technology is an innovative technology that has rapidly developed in recent years, combining spectral and image information to enable a new method of nondestructive testing [3]. This method may be used to get not just spectral information about the sample, but also to acquire a sample picture [4]. This fusion allows for more extensive and rich data and information while doing seed classification [5]. Hyperspectral imaging technology allows for the acquisition of spectral information at various wavelengths as well as sample morphological properties [6]. This improves the accuracy and efficiency of quality analysis. This technique is also widely utilized in agriculture for seed quality testing, disease diagnostics, and other applications, providing significant assistance for agricultural productivity [7]. Hyperspectral technology has grown in popularity in many industries, including agriculture, food, medicine, and industry, because of its numerous benefits, including simplicity, high accuracy, speedy identification, and low cost. The extensive usage of this technology is credited to its simplicity, high accuracy, rapid recognition speed, and low cost, making it a vital tool in these sectors [8]. Hyperspectral imaging has been utilized to categorize soybean cultivars in recent years due to its advantages of quick measurements and little non-destructive sample preparation [9].

In a recent overseas study, ZHOU Quan et al. (2021) [10] properly assessed the morphology of embryos and non-embryos of six maize seed types using hyperspectral image processing mixed with convolutional neural network (CNN) and partition voting methods. The study found that the accuracy rates were 97.7% and 98.15%, respectively. This suggests that combining hyperspectral technologies with deep learning approaches has yielded impressive results in seed variety categorization. WU Na(2021) [11] et al. used hyperspectral imagery and deep migratory learning to quickly and accurately classify four different agricultural seeds (rice, oats, wheat, and cotton) under restricted sample conditions. An accuracy of up to 99.57% was achieved. The findings demonstrated that combining hyperspectral imaging technology with a neural network may improve seed classification accuracy. Li Hao [12] (2021) et al. developed classical machine learning models (k-nearest neighbor, support vector machine, and partial least squares discriminant analysis) as well as a 1D CNN model using varying numbers of soybean sample sets. The 1D CNN model is the most stable and achieves classification accuracy of more than 98%. The models were assessed using five-fold cross-validation, and they obtained greater than 95% accuracy on both the training and validation sets. These approaches help identify and classify seeds.

According to literature reviews, some scholars and researchers have begun to investigate approaches for categorizing agricultural seeds using hyperspectral techniques. This research has mostly focused on the optimization of classification algorithms, with substantial progress being achieved. This suggests that hyperspectral technology has the potential to be used in agricultural seed classification research, particularly for classification algorithms that produce good results. The redundant bands of hyperspectral data greatly increase the number of subspace features, so this paper attempts to improve the accuracy of soybean seed classification in terms of both hyperspectral image reconstruction to generate a new dataset and algorithm optimization in soybean variety classification using relevant deep learning algorithms

borrowed from previous authors. It seeks an effective, non-destructive, and user-friendly approach for classifying soybean seed varieties.

## 2. Materials and methods

### 2.1. Sample preparation

The soybean samples utilized in this experiment were collected from Jilin Province's Academy of Agricultural Sciences, and they consisted of seven different varieties of soybean seeds, each comprising around 200 seeds. Experts picked the seeds, which were then meticulously washed by hand to eliminate any contaminants and dust [13]. The samples utilized in the studies were of remarkable quality, with no significant faults or damage, ensuring the dependability and precision of the experimental findings. Each species' samples were chosen to be more similar in size and saturation. Fig 1 shows the RGB photos of soybean seeds from the seven samples used in the experiment.

### 2.2 Hyperspectral Image acquisition

The experiment used the Resonon Benchtop Hyperspectral Imaging System, manufactured by Resonon Inc. in the United States, which included a Pika hyperspectral imager, a linear mobile platform, a mounting tower, an illumination device, and a software control system. The imaging spectrometer is the Pika XC2 (10.1cmx27.5cmx7.4cm), which has a wavelength range of 400nm-1000nm, a spectral resolution of 1.3nm, a sampling interval of 0.67nm, and an exposure duration of 56.37ms. Spectronon, a spectrum acquisition and analysis program, captures 462 spectral bands with a 1600×1000 picture resolution. The image resolution was 1600×1000, and the data files created after image capture were approximately 1.37GB apiece, stored in bil format.

When the hyperspectral images were collected, each species was placed as a research object on the electronically controlled displacement stage, and every effort was made to keep the placement uniform and at the same interval. As the platform moved, the camera scanned the entire platform. To avoid interference from other light in the room, the hyperspectral image acquisition process was done in a dark box, and the schematic diagram of the hyperspectral imager is shown in Fig 2.

### 2.3 Seed image segmentation

Firstly, the RGB images of a set of photographed samples are converted to grayscale maps, the binary image containing holes is obtained by binarizing the grayscale image, the seed shell matrix is obtained by eliminating the holes in the binary image and detecting the seed contours

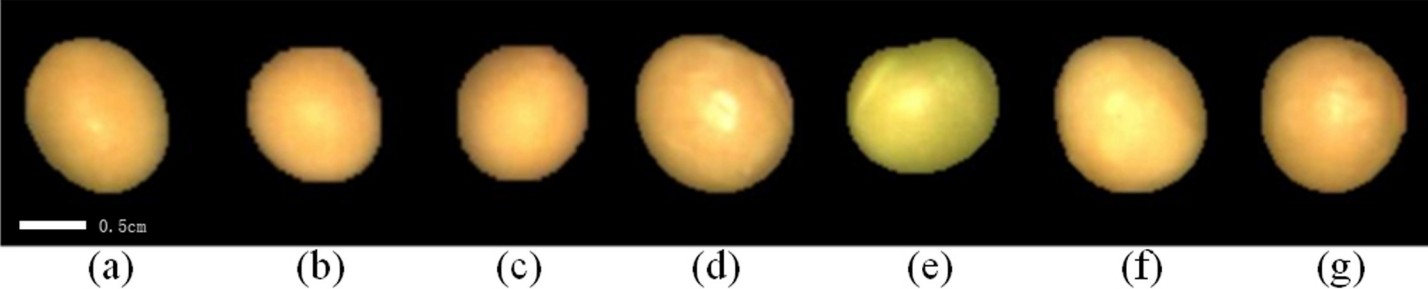

**Fig 1.** RGB images of Soybean seed grains: (a) H1, Scale bar:0.5cm (b) H2, Scale bar:0.5cm (c) H3, Scale bar:0.5cm (d) H4, Scale bar:0.5cm (e) H5, Scale bar:0.5cm (f) H6, Scale bar:0.5cm (g) H7, Scale bar:0.5cm.

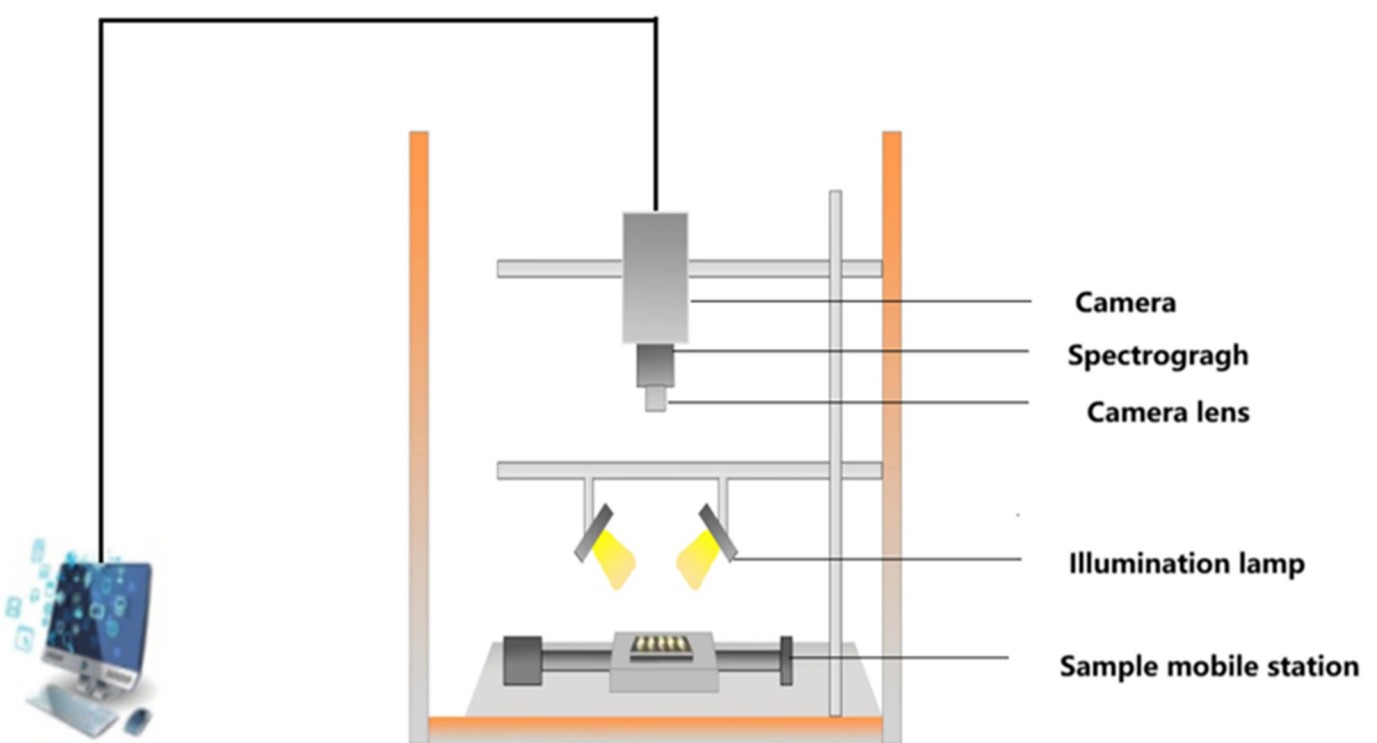

**Fig 2. Schematic diagram of the hyperspectral imager.**

using the unfolding operation, the segmentation mask achieves fine-grained segmentation of the image region by classifying and labeling each pixel, which in turn yields the image of a This method improves the accuracy of detecting soybean seed areas [14]. Fig 3 depicts the sequence of processes in image preprocessing.

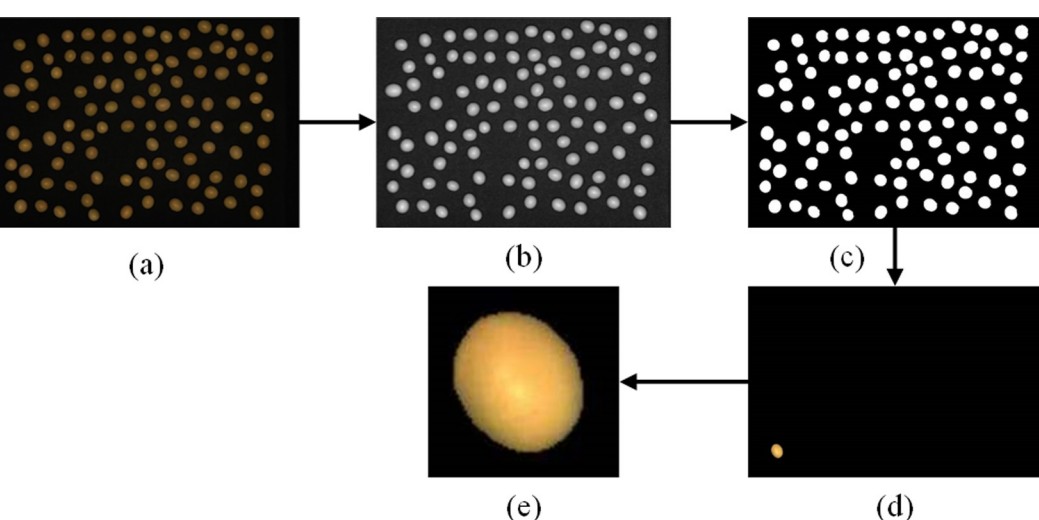

**Fig 3.** Image preprocessing: (a) a series of photographs; (b) grayscale map; (c) binari- zation map; (d) single seed mask map; (e) segmented single seed map;.

## 2.4 Data enhancement techniques

A large amount of training data helps prevent overfitting and improve model classification performance, hence data augmentation methods are frequently utilized to increase the data set. To account for the uncertainty of identified seed states in real-world scenarios, the training pictures were rotated by 45˚ [15], 90˚, color-enhanced, brightness-enhanced, mirrored, contrast-enhanced, and normalized. After eliminating imperfect or sticky soybean seeds, the overall number of seeds was increased by sixfold, from around 1050 to 7616, for seed segmentation. The augmented picture is trained alongside the sample image, improving the model's classification accuracy and resilience while also increasing its applicability [16]. Fig 4 displays extensive information.

## 2.5 Division of the data set

Deep learning algorithms (ResNet18、ResNet34、ResNet50、ResNet101、CBAM-ResNet34、SENet-ResNet34、SENet-ResNet34-DCN, seven residual network models) were used to classify seven types of soybean seeds for the study, and then to develop a fast and non-destructive variety recognition method based on soybean seed image data. The picture dataset was separated into three sets: training, validation, and test, in the ratio of 7:2:1. To develop the classification model, H1, H2, H3, H4, H5, H6, and H7 samples were gathered, as indicated in Table 1.

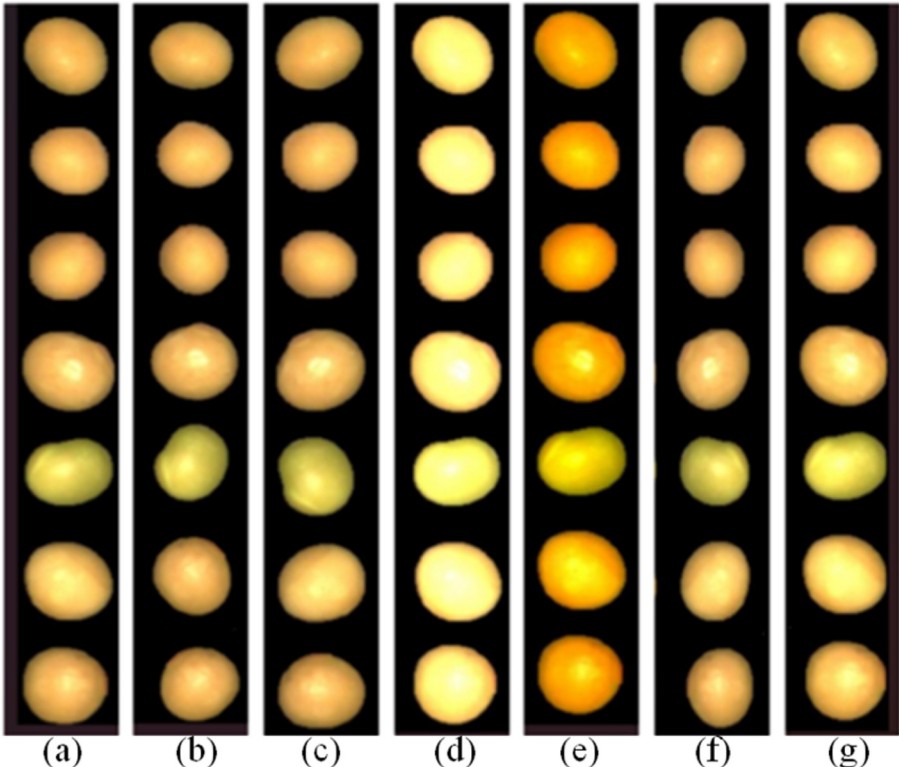

**Fig 4.** Dataset expansion: (a) Original image (b) Rotated 45˚ (c) Rotated 90˚ (d) Bright- ness enhancement (e) Color enhancement (f) Mirroring (g) Contrast enhancement.

**Table 1. Data for training and validation.**

| Name | Test Set | Training Set | Validation Set | Total |
|---|---|---|---|---|
| H1 | 109 | 759 | 217 | 1085 |
| H2 | 109 | 759 | 217 | 1085 |
| H3 | 110 | 769 | 220 | 1099 |
| H4 | 109 | 759 | 217 | 1085 |
| H5 | 107 | 744 | 213 | 1064 |
| H6 | 112 | 784 | 224 | 1120 |
| H7 | 108 | 754 | 216 | 1078 |
| Total | 764 | 5328 | 1524 | 7616 |

## 2.6 RGB reconstruction of hyperspectral images

Hyperspectral imaging combines spectral analysis and imaging techniques to get both spectrum and image data about the sample to be evaluated. Hyperspectral pictures are three-dimensional data structures, with a two-dimensional image and spectral data as the third dimension. The hyperspectral picture is made up of grayscale images at various wavelengths, with one grayscale image acquired for each wavelength. Each pixel point in the hyperspectral image comprises a spectral curve. Hyperspectral photos reveal a variety of information [17]. RGB picture synthesis is based on the human eye's color perception system. The human eye's optic cone cells are classified into three categories based on their ability to see red, green, and blue spectra. When light from these three spectra enters the optic cone cells at the proper strength and ratio, we see a distinct color. RGB picture synthesis simulates the experience of three spectra by using red, green, and blue channels. Each channel is a grayscale picture with the brightness information for the matching color [18]. The visual data from these three channels may be superimposed to reconstruct a color image. The fixed wavelength range of the three RGB channels is the following: the red band is 620.05–779.75 nm, band number 173–292; the green band is 491.19 nm-558.79 nm, band number 76–127; and the blue band is 421.29–448.94 nm, band number 23–44. In creating genuine color photos, It is essential to choose a certain band from these wavelength ranges and assign it to the RGB channel, and the resulting pictures are all true colors [19]. We use the R, G, and B channel approach for hyperspectral image reconstruction, and enhancing or weakening particular color channels may result in changes in color characteristics. Fig 5 demonstrates several phases in hyperspectral picture creation.

After executing hyperspectral image reconstruction, its clarity and colorfulness are richer and more realistic than the original, resulting in a greater classification recognition rate by the later algorithm. The single-band grayscale map is used to derive the three sets of R, G, and B feature bands with the highest classification accuracy using the ResNet34 algorithm, and then the RGB pseudo-color image is reconstructed to construct a new dataset, which is subsequently segmented, and each seed is extracted from the image, and then recognized and compared with the original dataset using ResNet34 to validate the validity of the hyperspectral image reconstruction method. In this experiment, nine combinations with greater classification accuracies than the original dataset are picked. The classification norm of each combination is depicted in Fig 6. The original dataset 188-120-60 has a classification accuracy of 88.87% and a precision of 89%, and the nine combinations have a 2% improvement in classification accuracy and precision. The 180-77-37 dataset has a classification accuracy of 90.18% and a precision of 90.43%, which is the least improvement. The 188-83-41 dataset has the highest classification accuracy of 91.75% and the highest precision of 91.71%, which is the most

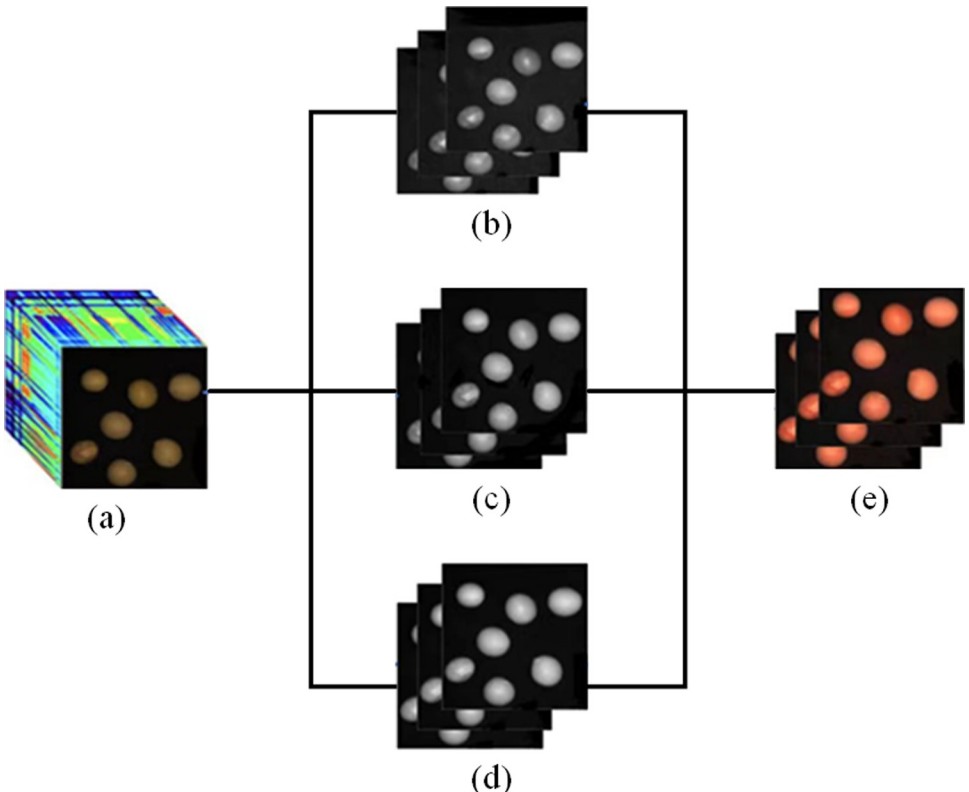

**Fig 5.** Hyperspectral reconfiguration (a) Raw hyperspectral image (b) Red band grayscale image (c) Blue band grayscale image (d) Green band grayscale image (e) reconstructed image.

improvement. 91.71%, with the most improvement, 2.88% and 2.71% over the original dataset. Therefore the 188-83-41 dataset is referred to as the new dataset and applied to the following algorithm for soybean seed classification.

## 2.7 Network models

When the depth of a CNN rises, gradient degradation and disappearance occur throughout the training process, resulting in convergence issues and low accuracy [20, 21]. The ResNet family was designed to solve the problem of gradient disappearance and gradient explosion in deep neural networks, but it degrades performance when training deep neural networks [22]. In this paper, we add the attention mechanism and replace some of the ordinary convolutions with deformable convolutions together to improve the overall performance and classification accuracy of soybean seeds.

**2.7.1 Squeeze-and-Excitation Networks.** SENet (Squeeze-and-Excitation Networks) attention mechanism module, which is a channel type of attention mechanism [23]. The main content is squeeze and excitation. On the basis of the original learning mechanism, open a new network path, after the operation, to obtain the degree of attention of each channel in the feature map, and according to this degree, to configure an attention weight for each feature channel, so that the convolutional network can pay more attention. Before implementing the SE attention mechanism, each channel of the feature map received the same level of attention; however, after passing through the SENet, different colours represent different weights, causing the importance of each feature channel to change, resulting in the network consciously

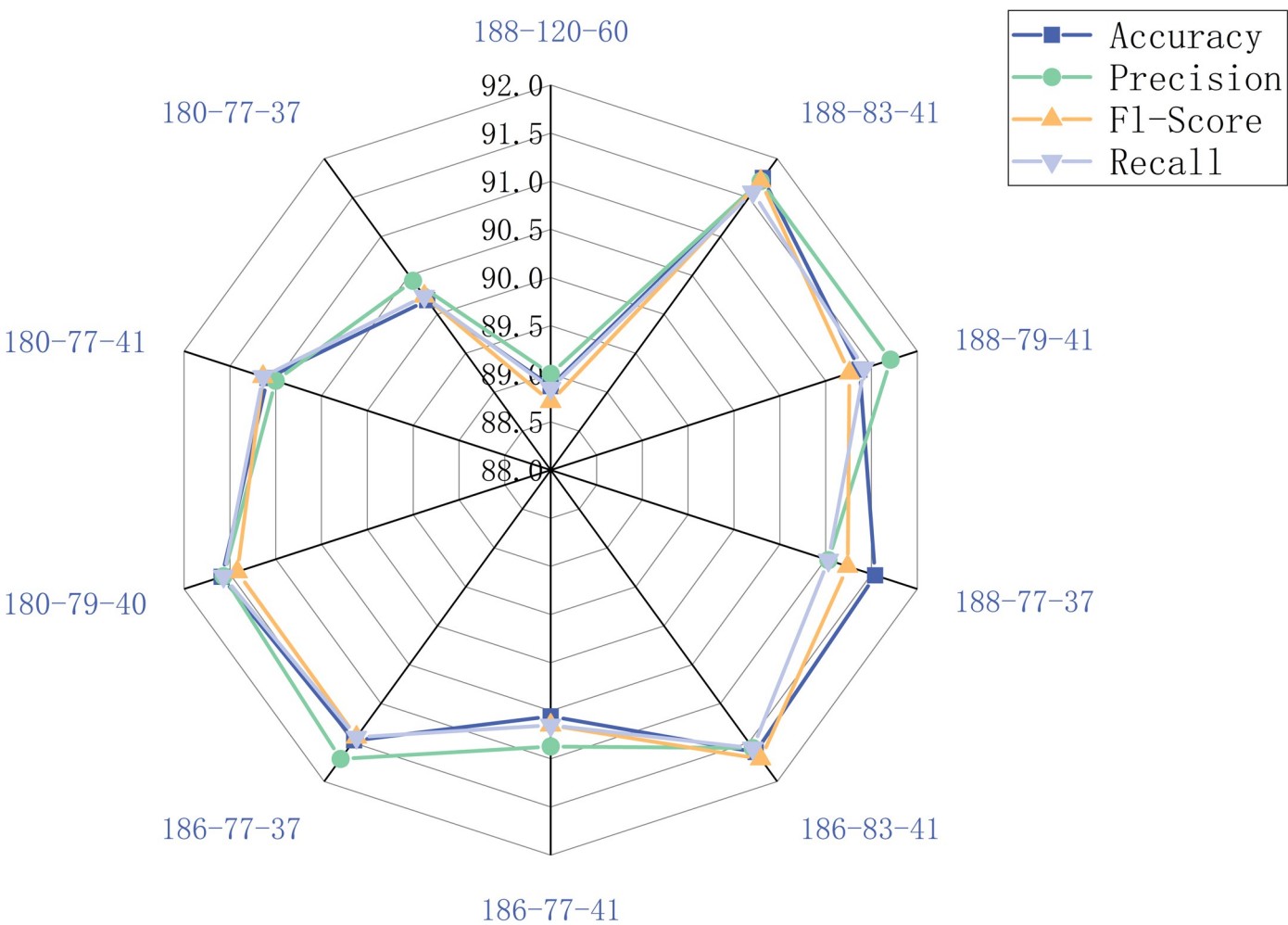

**Fig 6. Classification accuracy plot for each group of reconstructed images.**

focusing on certain channels with high weight values. As indicated in Fig 7. Because the maximum pooling layer typically loses some global information, it only focuses on the maximum value of each region. Adding the SE attention mechanism introduces global information by averaging the pooling and learned activation weights globally, improving the network's ability to perceive global features. By introducing the SE attention mechanism after Conv2 of ResNet34, the model better learns and generalises the features of the input image, thus improving the overall performance.

**2.7.2 Convolutional Block Attention Module.** CBAM (Convolutional Block Attention Module) consists of a channel attention mechanism (channel) and a spatial attention mechanism (spatial). CBAM begins with the two scopes of channel and spatial, introduces two analysis dimensions of spatial and channel attention, and realizes a channel to spatial sequential attention structure [24]. Spatial attention allows the neural network to pay more attention to the pixel regions in the image that play a decisive role in classification while ignoring the irrelevant regions, whereas channel attention is used to deal with the allocation relationship of the feature map channels, and simultaneous allocation of attention to the two dimensions improves the effect of the attention mechanism on model performance. The flow of CBAM is depicted in Fig 8. To begin, an intermediate feature map is provided as input to the Channel

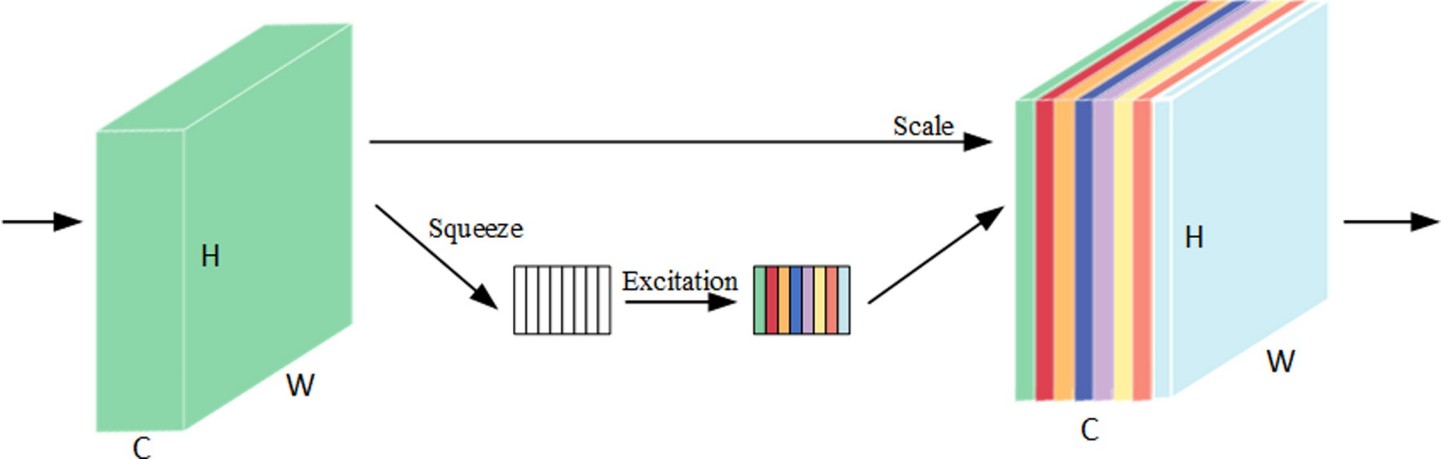

**Fig 7. SE module diagram.**

Attention Module, which then applies the attention weights to the intermediate feature map. The feature map with channel attention is then sent to the Spatial Attention Module to gain spatial attention, after which the attention weights are applied to the feature map. Following serial operation of these two Attention Modules, the original feature maps are processed by both channel and spatial attention processes, which adaptively refine the features. In this experiment, by adding CBAM to the maximum pooling layer of ResNet34 and after Conv2, the network is able to focus more on the key channel and spatial information in order to enhance the model's ability to model image features and thus improve classification performance.

**2.7.3 Deformable convolution.** Convolutional neural networks' capacity to replicate geometric changes is limited since its building pieces have a predetermined geometric structure. Thus, for good visual recognition, the sample location or deformation kernel must be adaptively chosen based on the object. In this research, we investigate using deformable convolutional layers (DCN) in the initial stage of the convolution of ResNet34's Conv2 instead of typical normal convolutional layers. The deformable convolutional layer utilized is seen in Fig 9. The deformable convolutional layer may adapt to the input data's non-uniform feature distribution by dynamically adjusting the shape of the convolution kernel to collect local characteristics. Second, as compared to the fixed shape convolutional kernel, the flexible convolutional layer is more expressive, which improves the model's capacity to learn

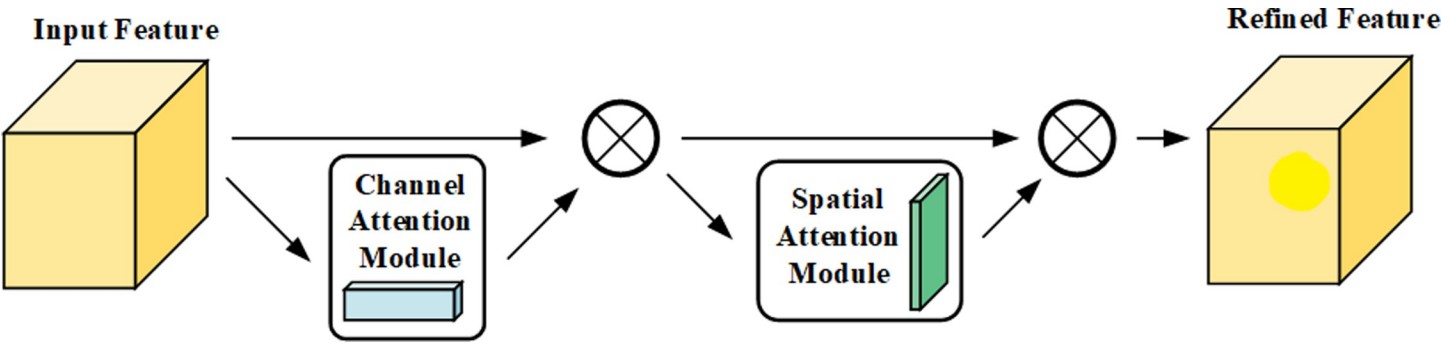

**Fig 8. Flow chart of CBAM.**

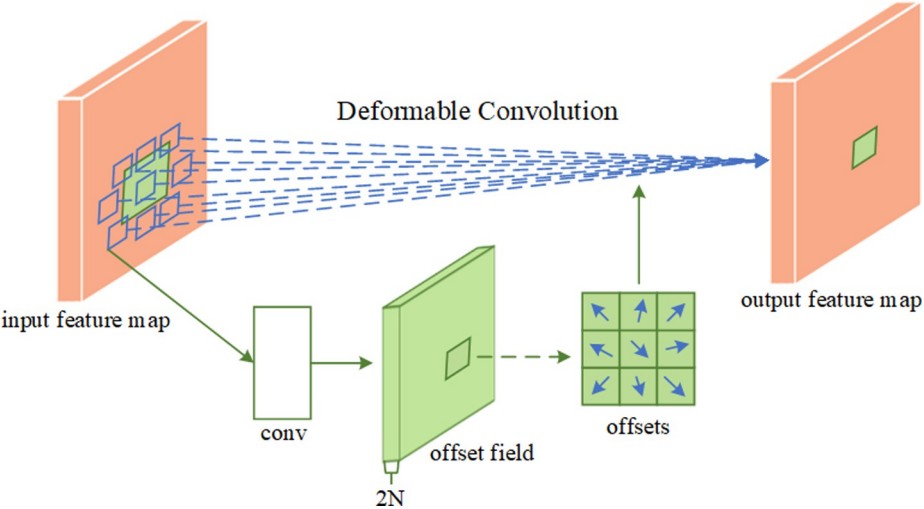

**Fig 9. Deformable convolutional layer.**

complicated patterns and abstractions. Furthermore, by using deformable convolutional layers, the network model is better able to manage spatial fluctuations in the input data, increasing the recognition accuracy of the target's various breathing activities. Overall, the introduction of deformable convolutional layers is predicted to improve the performance of convolutional neural networks in applications such as image processing, and we propose extending this concept to soybean seed classification.

**2.7.4 Proposed model.** This experiment's CBAM-ResNet34, SENet-ResNet34, and SENet-ResNet34-DCN models offer a novel way to categorize soybean seeds. The eight components of the SENet-ResNet34-DCN model's network architecture are two SENet Attention Mechanism Modules, five convolutional layers, and a fully connected layer at the end. The last layer is one that is entirely linked. The output of the convolutional layer is completed by using ReLU as the activation function after batch normalisation, which comes after the convolution procedure. Additionally, maximum pooling and average pooling algorithms are utilized to prevent overfitting and minimize the amount of parameters and computation in the network. To make the supplied picture 224 × 224 × 3, it is scaled. The output of the fully connected layer was input into Softmax based on the provided dataset, which produced a probability distribution that could be used to estimate the varieties of the seven soybean seeds. Fig 10 depicts the algorithm's flow for soybean seeds in the experiment's network.

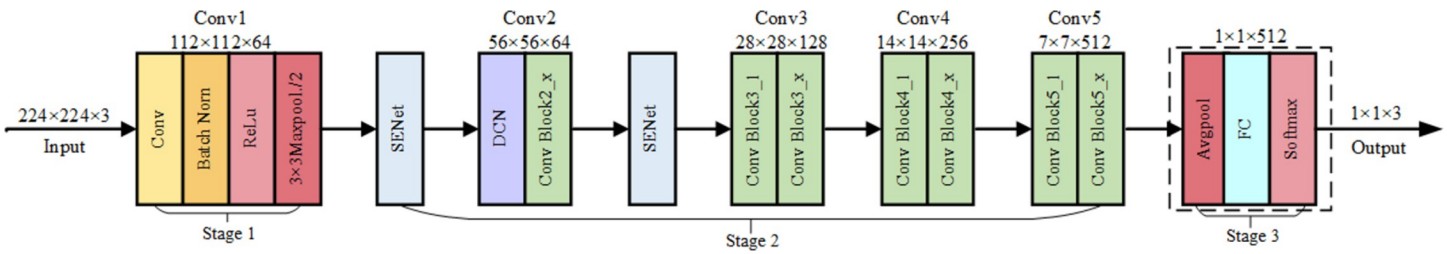

**Fig 10. Algorithm flowchart.**

## 2.8 Experimental procedures

The experiment was trained on 5328 photos, verified using 1524 images, and tested with 764 images. The H1, H2, H3, H4, H5, H6, and H7 variants are studied [25]. After the model has been trained, its performance is tested and compared using training and validation sets. The operating system is Windows 11, the CPU is a 12th Gen Intel(R) Core(TM) i9-12900K 3.20 GHz, the graphics card is an NVIDIA GeForce GTX 3090Ti, the Pycharm version is 2021 Community Edition, and the soybean seed categorization is constructed using the PyTorch deep learning framework model.

The model described in this experiment employs SGD as an optimizer, is trained with an initial learning rate of 0.001, and the network's loss function is lowered by changing the weights parameter. Epoch represents the full training cycle for the soybean seed dataset, with the greatest value corresponding to the minimal loss function's limiting value. The maximum number of training rounds is set to 50, with a minimum batch size of 8, momentum is 0.9, and weight_decay is 0.01. These options provide better outcomes in the optimizer. The categorization process of soybean seeds in the network participating in the experiment is depicted in Fig 11.

## 3. Results

The prepared datasets were trained using the ResNet18, ResNet34, ResNet50, ResNet101, CBAM-ResNet34, SENet-ResNet34, and SENet-ResNet34-DCN models applying the hyperparameters described above. All networks were trained for 50 epochs. The accuracy and loss of training and validation data for each epoch are shown in Fig 9. In the initial phase (1–10 epochs), the loss values decrease dramatically, but the accuracy increases significantly. Finally, the ResNet18 model achieves more than 65% accuracy in the training phase and the loss of the model is stable at around 0.7, while the remaining six models achieve more than 95% accuracy in the training phase and the loss of the model is stable at around 0.3, which indicates that the remaining six models are more stable and reliable. In addition, the SENet-ResNet34-DCN model achieved the convergence process in about 10 periods. As shown in Fig 12, after this

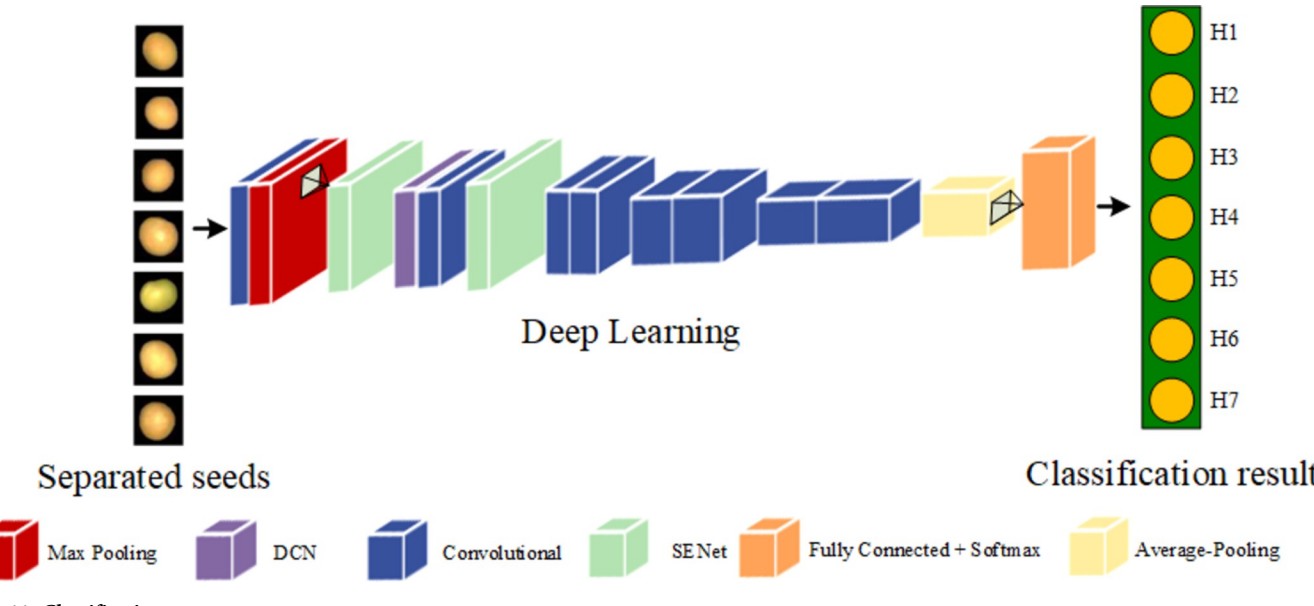

**Fig 11. Classification process.**

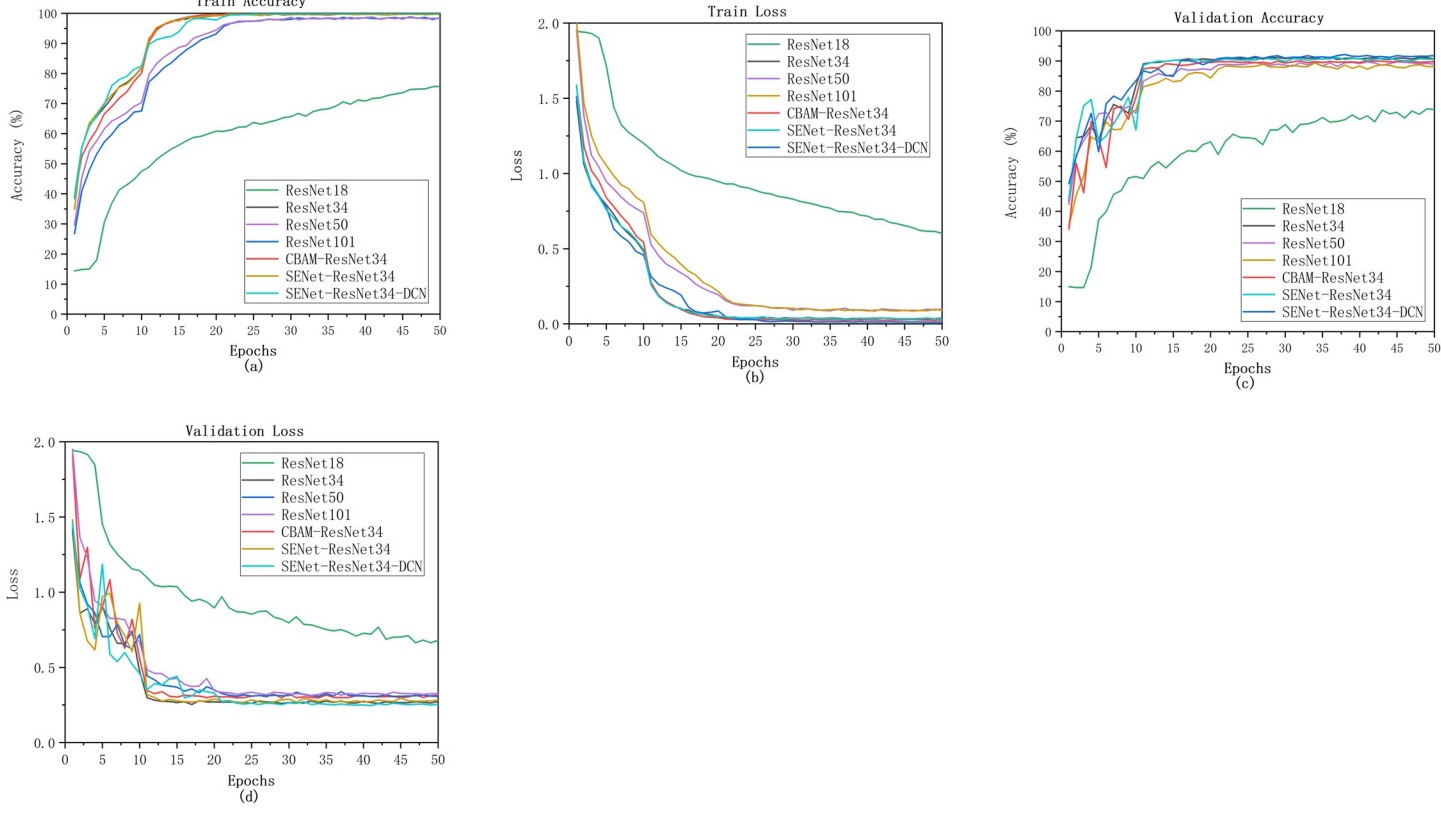

**Fig 12.** (a) Train accuracy (b) Train loss (c) Validation accuracy (d) Validation loss.

period, the validation set accuracy and loss curves flattened out and the difference between the accuracy and loss values of the validation and training data decreased. This result indicates that the performance of the SENet-ResNet34-DCN model is satisfactory for the final classification purpose.

Following training, a confusion matrix is generated for each classification technique, as illustrated in Fig 13. The data on the confusion matrix depicts both the actual categories in the sample and the categories predicted by the classifier. These four measures are often true positives (TP), true negatives (TN), false positives (FP), and false negatives (FN) [26]. In this article, TP and TN represent the right identification of maize seeds, whereas FP and FN represent the wrong identification. The models' performance is assessed using statistical metrics from the confusion matrix, such as accuracy, precision, specificity, recall, and F1-score [27], which can be found in Table 2 and Fig 14.

Every model in this experiment was able to identify seven different types of soybean seeds. With an accuracy of 94.24%, SENet-ResNet34-DCN leads the group, while ResNet18 ranks last with 72.25%. The precision of SENet-ResNet34-DCN is 94.14%, which is 2.28% better than that of ResNet34. The total test set times of ResNet18, ResNet34, ResNet50, ResNet101, CBAM-ResNet34, SENet-ResNet34, and SENet-ResNet34-DCN models are 3.41s, 4.05s, 3.37s, 3.75s, 2.95s, 2.83 s, 2.77s, and the SENet-ResNet34-DCN model has the shortest time, which is reduced by 0.06s-1.28s, and the test times for a single image are 0.0045s, 0.0053s, 0.0044s, 0.0049s, 0.0038s, 0.0037s, and 0.0036s, respectively. The SENet-ResNet34-DCN model reduced it by 0.0001s-0.0023s.Furthermore, the classification outcomes and confusion matrix show how well SENet-ResNet34-DCN performs. By improving the model's emphasis on significant

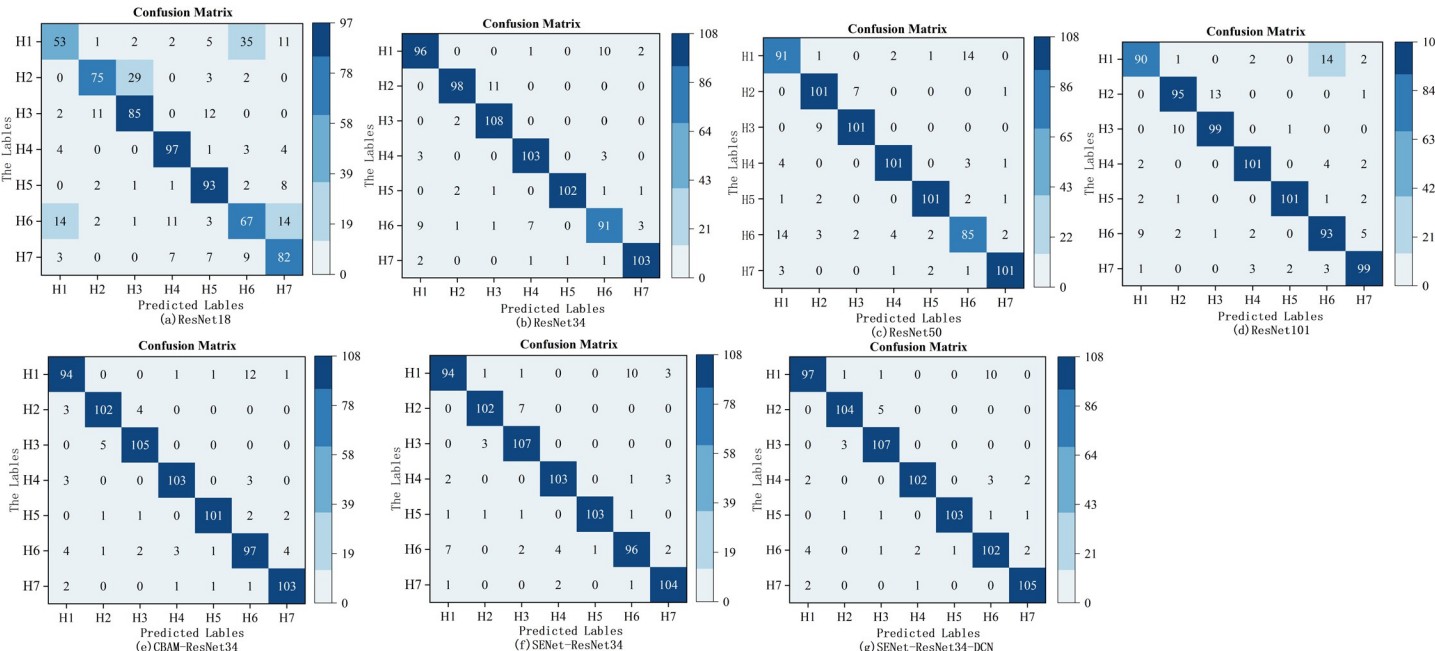

**Fig 13.** Confusion matrix (math.) (a) ResNet18 (b) ResNet34 (c) ResNet50 (d) ResNet101 (e) CBAM-ResNet34 (f) SENet-ResNet34 (g) SENet-ResNet34-DCN.

characteristics and improving its ability to capture local information, the experimental findings further demonstrate that the network offers a strong basis for the successful completion of following classification tasks.

The SENet-ResNet34-DCN suggested in this work has comparatively greater performance and accuracy, as seen in Fig 14, and other metrics are similarly superior than previous models. Because of its superior classification findings, the usage of the SENet-ResNet34-DCN model in this investigation validates the viability of the ResNet34-based improvement in this work.

The results of classifying all soybean varieties' seeds under various models are displayed in Table 3 and Fig 15. The figure illustrates that, for seven soybean varieties, ResNet18 has a low classification accuracy, while the other six models have a classification accuracy of more than 75%.For the H1, H2, H3, H4, H5, H6, and H7, the SENet-ResNet34-DCN network model has the best classification accuracy, with scores of 88.99%, 95.41%, 97.27%, 93.57%, and 96.26%, 91.07%, 97.22%, respectively. H1 and H6, on the other hand, perform worse in classification than the other models, with the lowest values of 48.62% and 59.82%, respectively. The results indicate that H1 and H6 may overlap with the other five varieties in terms of features and have poor differentiation.

**Table 2. Classification results of models.**

| Name | Accuracy(%) | Specificity (%) | Recall(%) | Precision(%) | F1-Score(%) | Time(s) |
|---|---|---|---|---|---|---|
| ResNet18 | 72.25 | 95.69 | 72.33 | 72.43 | 71.91 | 0.0045 |
| ResNet34 | 91.75 | 98.62 | 91.80 | 91.86 | 91.77 | 0.0053 |
| ResNet50 | 89 | 98.18 | 89.28 | 75.71 | 89.14 | 0.0044 |
| ResNet101 | 88.48 | 98 | 88.85 | 89 | 88.85 | 0.0049 |
| CBAM-ResNet34 | 92.28 | 98.71 | 92.35 | 92.35 | 92.32 | 0.0038 |
| SE-ResNet34 | 92.80 | 98.42 | 92.71 | 92.85 | 92.85 | 0.0037 |
| SENet-ResNet34-DCN | 94.24 | 99.03 | 94.14 | 94.14 | 94.14 | 0.0036 |

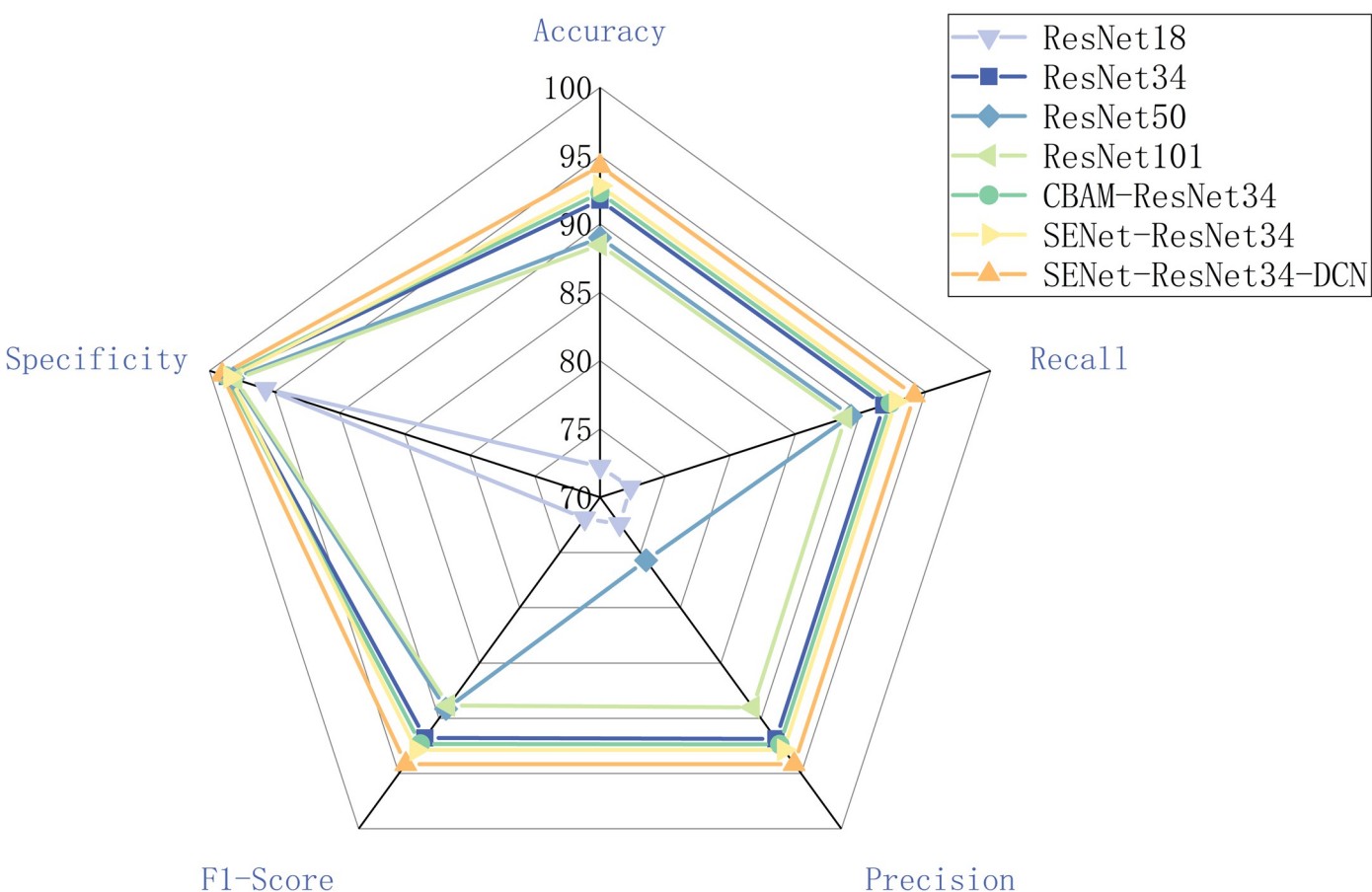

**Fig 14. Classification results of models.**

## 4. Discussion

For the non-destructive, quick, and effective classification and identification of soybean seeds, this study used residual network models (ResNet18, ResNet34, ResNet50, ResNet101, CBAM--ResNet34, SENet-ResNet34, and SENet-ResNet34-DCN). Their performance was assessed and compared, and the tested accuracy of the SENet-ResNet34-DCN network model was 94.24%. The study demonstrates the excellent accuracy and stability of SENet-ResNet34-DCN in the job of classifying and recognizing soybean seeds. Richer color information may be gained in

**Table 3. Statistics of classification results for all varieties.**

| Model | Classification Accuracy(%) | | | | | | |
|---|---|---|---|---|---|---|---|
| | **H1** | **H2** | **H3** | **H4** | **H5** | **H6** | **H7** |
| ResNet18 | 48.62 | 68.81 | 77.27 | 89.99 | 86.92 | 59.82 | 75.93 |
| ResNet34 | 88.07 | 89.91 | 98.18 | 94.5 | 95.33 | 81.25 | 95.37 |
| ResNet50 | 83.48 | 92.66 | 91.81 | 92.66 | 94.39 | 75.89 | 93.51 |
| ResNet101 | 82.56 | 87.15 | 90 | 92.66 | 94.39 | 83.03 | 91.66 |
| CBAM-ResNet34 | 86.24 | 93.58 | 95.45 | 94.5 | 94.39 | 86.61 | 95.37 |
| SE-ResNet 34 | 86.23 | 93.57 | 97.27 | 94.49 | 96.26 | 85.71 | 96.29 |
| SENet-ResNet34-DCN | 88.99 | 95.41 | 97.27 | 93.57 | 96.26 | 91.07 | 97.22 |

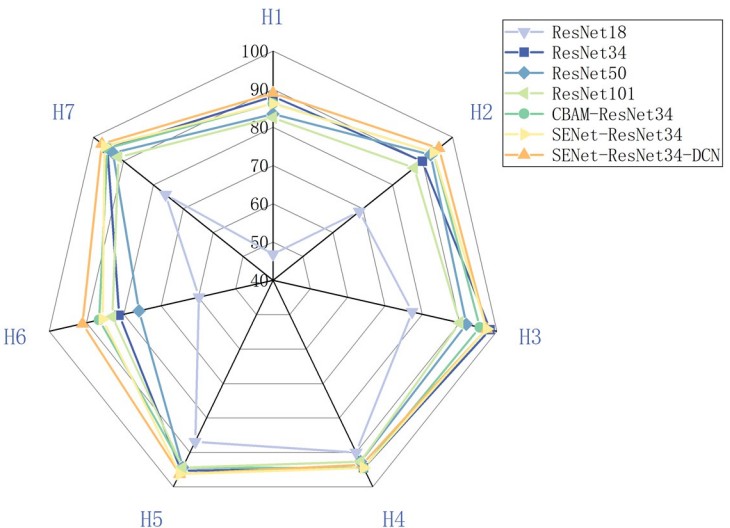

**Fig 15. Classification results for seed.**

this study's dataset by applying the hyperspectral image reconstruction approach, which also increases the accuracy of soybean seed classification from 88.87% to 91.75%. Feature band selection methods such as SPA in other literatures are based on spectral angles, and for full or hyperspectral data, a large amount of band information needs to be processed, and manually analyzing the spectral properties of each band can become very time-consuming and complex. The resulting bands for image reconstruction cannot improve the classification accuracy and other metrics of soybean seeds. The RGB reconstruction method proposed in this paper, which is based on the image perspective, uses the principle of three basic colors to change the color and texture characteristics of the image, and filters out some important bands before reconstruction, and the resulting new dataset has more obvious characteristics, and the classification accuracy of the nine datasets in the experiments and other metrics have been improved [28]. To guarantee that high-quality seeds are accessible for agricultural production, the SENet-ResNet34-DCN model may be tried to be applied to various areas and seed kinds in terms of categorization. In addition, the method has the potential to be applied in seed variety identification, and the developed variety classification model can be applied to seed sorting machinery to prevent counterfeit seeds from disrupting the market and ensure food security [29, 30]. However, the size of the dataset in this study is relatively small, and further expansion and exploration of other varieties and generalisation capabilities are required [31].

## Author Contributions

**Conceptualization:** Xu Yang, Kejia Ma.

**Data curation:** Xu Yang.

**Formal analysis:** Xu Yang.

**Funding acquisition:** Shaozhong Song.

**Investigation:** Kejia Ma.

**Methodology:** Xu Yang.

**Project administration:** Shaozhong Song.

**Resources:** Shaozhong Song.

**Software:** Xu Yang.

**Supervision:** Shaozhong Song.

**Validation:** Xu Yang, Kejia Ma, Dejia Zhang, Xiaofeng An.

**Visualization:** Kejia Ma.

**Writing – original draft:** Kejia Ma, Xiaofeng An.

**Writing – review & editing:** Shaozhong Song.

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
