## [Decision Letter · Decision Letter 0]

11 Jun 2024

PONE-D-24-15624Classification of soybean seeds based on RGB reconstruction of hyperspectral imagesPLOS ONE

Dear Dr. Song,

Thank you for submitting your manuscript to PLOS ONE. After careful consideration, we feel that it has merit but does not fully meet PLOS ONE’s publication criteria as it currently stands. Therefore, we invite you to submit a revised version of the manuscript that addresses the points raised during the review process.

We look forward to receiving your revised manuscript.

Kind regards,

Narendra Khatri, Ph.D.

Academic Editor

PLOS ONE

Journal Requirements:

'1.the Natural Science Foundation of Jilin Province (No.2020122348JC)

2.Innovation Capacity Project on Development and Reform Commission of Jilin Province（2020C019-6）'

Please state what role the funders took in the study.  If the funders had no role, please state: 'The funders had no role in study design, data collection and analysis, decision to publish, or preparation of the manuscript.' 

'The authors declare that they have no known competing financial interests or personal relationships that could have appeared to influence the work reported in this paper.'

Please complete your Competing Interests on the online submission form to state any Competing Interests. If you have no competing interests, please state 'The authors have declared that no competing interests exist.', as detailed online in our guide for authors at http://journals.plos.org/plosone/s/submit-now 

5. Thank you for uploading your study's underlying data set. Unfortunately, the repository you have noted in your Data Availability statement does not qualify as an acceptable data repository according to PLOS's standards.

Additional Editor Comments:

Major Revision for further consideration and review

Reviewers' comments:

Reviewer's Responses to Questions

**Comments to the Author**

1. Is the manuscript technically sound, and do the data support the conclusions?

Reviewer #1: Yes

Reviewer #2: Yes

2. Has the statistical analysis been performed appropriately and rigorously? 

Reviewer #1: Yes

Reviewer #2: Yes

3. Have the authors made all data underlying the findings in their manuscript fully available?

Reviewer #1: Yes

Reviewer #2: Yes

4. Is the manuscript presented in an intelligible fashion and written in standard English?

Reviewer #1: Yes

Reviewer #2: Yes

5. Review Comments to the Author

Reviewer #1: This study discusses the development of a hyperspectral imaging approach for classifying soybean seeds that uses hyperspectral RGB picture reconstruction. To this end, they collected hyperspectral images of seven varieties of soybean through hyperspectral imager, and by employing the principle of the three base colors, the R, G and B bands were selected. This was complemented by reconstructing the images with different texture and color characteristics to generate a new dataset for seed segmentation. This was concluded by comparing the classification effect of the seven models. Experimental assessments supported the claim and the authors tried to prove the proposed approach with different set of analyses. While the work is comprehensive and supported by extensive data, there are some works in the literature with analogous methods that have not been discussed in the current manuscript, such as: Sensors 2019 Dec; 19(23): 5225, Front Plant Sci. 2022; 13: 1098864, and many other works can be found easily through a quick literature search. The authors should argue the novelty of the work compared to recent studies and show what parameters have been optimized and how the proposed technique addresses the current limitations compared to those studies. Here are some additional comments:

1) The provided images need scalebars.

2) What is the sample to result duration compared to other technologies?

3) The precision of the proposed system should be discussed further and evaluated quantitatively.

Reviewer #2: Overall, the quality of the paper is good. The dataset samples could be considered more for evaluation purposes. The analysis of the results could be improved. The extensive literature review has been made.

6. PLOS authors have the option to publish the peer review history of their article (what does this mean?). If published, this will include your full peer review and any attached files.

Reviewer #1: No

Reviewer #2: No

---

## [Author Response · Author response to Decision Letter 0]

26 Jun 2024

Reviewer #1: This study discusses the development of a hyperspectral imaging approach for classifying soybean seeds that uses hyperspectral RGB picture reconstruction. To this end, they collected hyperspectral images of seven varieties of soybean through hyperspectral imager, and by employing the principle of the three base colors, the R, G and B bands were selected. This was complemented by reconstructing the images with different texture and color characteristics to generate a new dataset for seed segmentation. This was concluded by comparing the classification effect of the seven models. Experimental assessments supported the claim and the authors tried to prove the proposed approach with different set of analyses. While the work is comprehensive and supported by extensive data, there are some works in the literature with analogous methods that have not been discussed in the current manuscript, such as: Sensors 2019 Dec; 19(23): 5225, Front Plant Sci. 2022; 13: 1098864, and many other works can be found easily through a quick literature search. The authors should argue the novelty of the work compared to recent studies and show what parameters have been optimized and how the proposed technique addresses the current limitations compared to those studies.

The addition reads: Feature band selection methods such as SPA in other literatures are based on spectral angles, and for full or hyperspectral data, a large amount of band information needs to be processed, and manually analyzing the spectral properties of each band can become very time-consuming and complex. The resulting bands for image reconstruction cannot improve the classification accuracy and other metrics of soybean seeds. The RGB reconstruction method proposed in this paper, which is based on the image perspective, uses the principle of three basic colors to change the color and texture characteristics of the image, and filters out some important bands before reconstruction, and the resulting new dataset has more obvious characteristics, and the classification accuracy of the nine datasets in the experiments and other metrics have been improved [28].

1) The provided images need scalebars.

Response: The actual diameter of a soybean is around 0.5cm, and the diameter of the soybean seed in the paper is around 1cm, magnified twice, with a scale of 0.5cm. Specific changes have been made to Figure 1 in the thesis by adding a scale label and changing the legend to a scale of 0.5 cm.

2) What is the sample to result duration compared to other technologies?

The addition reads: The total test set times of ResNet18, ResNet34, ResNet50, ResNet101, CBAM-ResNet34, SENet-ResNet34, and SENet-ResNet34-DCN models are 3.41s, 4.05s, 3.37s, 3.75s, 2.95s, 2.83 s, 2.77s, and the SENet-ResNet34-DCN model has the shortest time, which is reduced by 0.06s-1.28s, and the test times for a single image are 0.0045s, 0.0053s, 0.0044s, 0.0049s, 0.0038s, 0.0037s, and 0.0036s, respectively. The SENet-ResNet34-DCN model reduced it by 0.0001s-0.0023s.

3) The precision of the proposed system should be discussed further and evaluated quantitatively.

The addition reads: The original dataset 188-120-60 has a classification accuracy of 88.87% and a precision of 89%, and the nine combinations have a 2% improvement in classification accuracy and precision.The 180-77-37 dataset has a classification accuracy of 90.18% and a precision of 90.43%, which is the least improvement.The 188-83-41 dataset has the highest classification accuracy of 91.75% and the highest precision of 91.71%, which is the most improvement. 91.71%, with the most improvement, 2.88% and 2.71% over the original dataset. Therefore the 188-83-41 dataset is referred to as the new dataset and applied to the following algorithm for soybean seed classification.( 222 to 230)

The precision of SENet-ResNet34-DCN is 94.14%, which is 2.28% better than that of ResNet34.(361-362)

Reviewer #2: Overall, the quality of the paper is good. The dataset samples could be considered more for evaluation purposes. The analysis of the results could be improved. The extensive literature review has been made.

The addition reads: Feature band selection methods such as SPA in other literatures are based on spectral angles, and for full or hyperspectral data, a large amount of band information needs to be processed, and manually analyzing the spectral properties of each band can become very time-consuming and complex. The resulting bands for image reconstruction cannot improve the classification accuracy and other metrics of soybean seeds. The RGB reconstruction method proposed in this paper, which is based on the image perspective, uses the principle of three basic colors to change the color and texture characteristics of the image, and filters out some important bands before reconstruction, and the resulting new dataset has more obvious characteristics, and the classification accuracy of the nine datasets in the experiments and other metrics have been improved [28].

---

## [Decision Letter · Decision Letter 1]

3 Jul 2024

Classification of soybean seeds based on RGB reconstruction of hyperspectral images

PONE-D-24-15624R1

Dear Dr. Song,

We’re pleased to inform you that your manuscript has been judged scientifically suitable for publication and will be formally accepted for publication once it meets all outstanding technical requirements.

Kind regards,

Narendra Khatri, Ph.D.

Academic Editor

PLOS ONE

Additional Editor Comments (optional):

Accept

---

## [Editor Report · Acceptance letter]

23 Aug 2024

PONE-D-24-15624R1 

PLOS ONE

Dear Dr. Song, 

I'm pleased to inform you that your manuscript has been deemed suitable for publication in PLOS ONE. Congratulations! Your manuscript is now being handed over to our production team.

Kind regards, 

on behalf of

Dr. Narendra Khatri 

Academic Editor

PLOS ONE